# Comparison of Survival Outcomes between Radical Antegrade Modular Pancreatosplenectomy and Conventional Distal Pancreatosplenectomy for Pancreatic Body and Tail Cancer: Korean Multicenter Propensity Score Match Analysis

**DOI:** 10.3390/cancers16081546

**Published:** 2024-04-18

**Authors:** Jaewoo Kwon, Huisong Lee, Hongbeom Kim, Sung Hyun Kim, Jae Do Yang, Woohyung Lee, Jun Suh Lee, Sang Hyun Shin, Hee Joon Kim

**Affiliations:** 1Department of Surgery, Kangbuk Samsung Hospital, Sungkyunkwan University School of Medicine, Seoul 06351, Republic of Korea; jaewoo8208.kwon@samsung.com; 2Department of Surgery, Ewha Womans University Mokdong Hospital, Seoul 07985, Republic of Korea; huisong.lee@ewha.ac.kr; 3Division of Hepatobiliary-Pancreatic Surgery, Department of Surgery, Samsung Medical Center, Sungkyunkwan University School of Medicine, Seoul 06351, Republic of Korea; hongbeom.kim@samsung.com; 4Department of Hepatobiliary and Pancreatic Surgery, Severance Hospital, Yonsei University College of Medicine, Seoul 03722, Republic of Korea; ohliebe@yuhs.ac; 5Department of Surgery, Jeonbuk National University Hospital, Jeonbuk National University Medical School, Jeonju 54896, Republic of Korea; yjd@jbnu.ac.kr; 6Division of Hepatobiliary and Pancreatic Surgery, Department of Surgery, Asan Medical Center, University of Ulsan College of Medicine, Seoul 05505, Republic of Korea; 7Department of Surgery, College of Medicine, Incheon St. Mary’s Hospital, Catholic University of Korea, Seoul 06591, Republic of Korea; rudestock@catholic.ac.kr; 8Division of Hepato-Pancreato-Biliary Surgery, Department of Surgery, Chonnam National University Medical School, Chonnam National University Hospital, Gwangju 61469, Republic of Korea

**Keywords:** pancreatic neoplasm, pancreatectomy, survival, prognosis

## Abstract

**Simple Summary:**

Our study aimed to compare the operative and survival outcomes between radical antegrade modular pancreatosplenectomy (RAMPS) and conventional distal pancreatosplenctomy (cDPS), and identify prognostic factors for left-sided pancreatic cancer. We performed a retrospective propensity score match (PSM) analysis from 333 patients who underwent RAMPS or cDPS for left-sided pancreatic cancer. After PSM, 99 cohorts were matched in each group. We compared survival and operative outcomes and assessed prognostic factors. R0 resection rate was similar between both groups, and R1 resection rate was a significant prognostic factor. RAMPS was found to be safe, feasible, and to improve the number of retrieved lymph nodes. However, when R0 resection was similar in both groups, RAMPS was not associated with improved survival. Adjuvant treatment was a significant independent prognostic factor for overall and disease-free survival, but operation type was not.

**Abstract:**

(1) Background: The aim of this study was to compare the survival benefit of radical antegrade modular pancreatosplenectomy (RAMPS) with conventional distal pancreatosplenectomy (cDPS) in left-sided pancreatic cancer. (2) Methods: A retrospective propensity score matching (PSM) analysis was conducted on 333 patients who underwent RAMPS or cDPS for left-sided pancreatic cancer at four tertiary cancer centers. The study assessed prognostic factors and compared survival and operative outcomes. (3) Results: After PSM, 99 patients were matched in each group. RAMPS resulted in a higher retrieved lymph node count than cDPS (15.0 vs. 10.0, *p* < 0.001). No significant differences were observed between the two groups in terms of R0 resection rate, blood loss, hospital stay, or morbidity. The 5-year overall survival rate was similar in both groups (cDPS vs. RAMPS, 44.4% vs. 45.2%, *p* = 0.853), and disease-free survival was also comparable. Multivariate analysis revealed that ASA score, preoperative CA19-9, histologic differentiation, R1 resection, adjuvant treatment, and lymphovascular invasion were significant prognostic factors for overall survival. Preoperative CA19-9, histologic differentiation, T-stage, adjuvant treatment, and lymphovascular invasion were independent significant prognostic factors for disease-free survival. (4) Conclusions: Although RAMPS resulted in a higher retrieved lymph node count, survival outcomes were not different between the two groups. RAMPS was a surgical option to achieve R0 resection rather than a standard procedure.

## 1. Introduction

Pancreatic ductal adenocarcinoma (PDAC) has a poor prognosis, with margin status and lymph node metastasis being recognized as unfavorable prognostic factors [1,2,3,4,5,6]. To address this issue, Strasberg et al. developed the concept of radical antegrade modular pancreatosplenectomy (RAMPS) [7]. RAMPS is a modified type of distal pancreatectomy to facilitate complete dissection of the N1 and N2 lymph nodes and obtaining negative posterior margins. RAMPS has been shown to be effective in achieving a higher lymph node yield and a negative posterior tangential margin in advanced left-sided PDAC when the tumor involves the peripancreatic soft tissue or peripancreatic lymph node metastases are suspected. However, it remains unclear whether RAMPS should be considered a standard operation for all left-sided PDACs, including non-advanced cases, or if it should be reserved as one of the surgical options only for advanced PDACs. Reports have shown that RAMPS has a 91% success rate in achieving negative tangential margins in patients [8]. However, the benefits of RAMPS over conventional distal pancreatosplenectomy (cDPS) in terms of survival outcomes have not been clearly established in previous studies [3,9,10,11]. This study aims to compare the operative and survival outcomes of RAMPS and cDPS to evaluate the value of RAMPS as a standard procedure in left-sided PDACs, as well as identify prognostic factors for left-sided pancreatic cancer.

## 2. Materials and Methods

### 2.1. Data Collection

We conducted a retrospective study of patients with PDAC who underwent RAMPS or cDPS at four tertiary hospitals between January 2010 and December 2020. RAMPS was defined according to the method described by Strasberg et al. [7]. The resection level at the pancreatic neck was reviewed from a postoperative CT scan and the extent of retropancreatic dissection and lymph node dissection was confirmed from the operation record. cDPS was defined as a distal pancreatectomy that does not fulfil the criteria for RAMPS (neck level transection, N1 and N2 lymph node dissection, and posterior dissection plane including the Gerota’s fascia). Cases in which the pancreatic resection was performed more distal than the pancreatic neck, depending on the location of the tumor, or a reduced lymph node dissection was performed according to the surgeon’s decision, or the posterior dissection plane did not follow the RAMPS plane were classified as cDPS. For patients with metastatic lesions, non-curative resection was excluded. Patients who underwent extended surgery such as combined organ resection or celiac axis resection that could not be classified in routine RAMPS nor cDPS were also excluded. We collected demographic and operative data, including age, sex, American Society of Anesthesiologist (ASA) score, body mass index (BMI), preoperative carcinoembryonic antigen (CEA) level, preoperative carbohydrate antigen 19-9 (CA19-9) level, neoadjuvant treatment, adjuvant treatment, operation type (RAMPS vs. cDPS), operation time, estimated blood loss, and hospital stay. Pathological data included tumor size, histologic differentiation, retrieved lymph node count, metastatic lymph node count, perineural invasion (PNI), lymphovascular invasion (LVI), and margin status. R1 resection was defined as the safety margin from the tumor to the resection margin being less than 1 mm. All postoperative morbidities were recorded and graded according to the Clavien–Dindo classification, and postoperative pancreatic fistula (POPF), post-pancreatectomy hemorrhage (PPH), and delayed gastric emptying (DGE) were graded according to the International Study Group of Pancreatic Surgery.

### 2.2. Statistical Analysis

A retrospective 1:1 propensity score matching analysis was conducted to compare the clinical and pathological outcomes of patients who underwent RAMPS or cDPS for PDAC. Six preoperative covariates, including age, sex, ASA score, tumor location, tumor size, and preoperative serum CA19-9 level, were used for propensity score matching. Because the tumor location influenced the decision regarding the pancreatic resection level and extent of lymph node dissection, tumor location was included in the propensity score matching. The matching was performed using a nearest neighborhood method with 0.01 of caliper. Continuous variables were compared using either the independent samples *t*-test or Mann–Whitney U test based on the normality test results, and categorical variables were compared using the chi-square test or Fisher’s exact test. The Kaplan–Meier method was used to estimate overall survival (OS) and disease-free survival (DFS), and the log-rank test was used to identify risk factors. Multivariate analysis was conducted using a Cox regression hazard model. Statistical significance was set at *p* < 0.05, and all statistical analyses were performed using SPSS version 26.0 (IBM SPSS Statistics for Windows, IBM Corp., Armonk, NY, USA).

## 3. Results

### 3.1. Baseline Characteristics before and after Propensity Score Matching

In this study, 333 patients with pancreatic ductal adenocarcinoma were initially enrolled. To balance the baseline characteristics between the two groups, 1:1 propensity score matching was performed, resulting in 99 patients included in both the RAMPS and cDPS groups. Table 1 demonstrates the baseline characteristics of the patients before and after PSM. After matching, both groups were found to be well balanced in terms of their preoperative characteristics.

### 3.2. Operative Outcomes

The results revealed a significantly higher retrieved lymph node count in the RAMPS group than in the cDPS group. However, the R0 resection rate was comparable between the two groups. Moreover, no significant differences were observed in the length of hospital stay, operation time, estimated blood loss, transfusion rate, and morbidity between the two groups. There was no mortality within 30 and 90 days. The recurrence rate and recurrence pattern were also found to be comparable between both groups. The detailed operative outcomes are presented in Table 2.

### 3.3. Comparison of Pathologic Results

No significant differences were found in terms of histologic differentiation, T-stage, perineural invasion, and lymphovascular invasion between the two groups. However, there was a significant difference in the number of positive lymph nodes and N-stage, with higher counts observed in the RAMPS group compared to the cDPS group. This observation raises the possibility of under-staging in the cDPS group. The detailed pathologic results can be found in Table 3.

### 3.4. Comparison of Survival Outcomes

The study included a mean and median follow-up time of 30.84 and 22.5 months (range 3–133 months) for all matched patients, with no significant differences in the mean and median follow-up times between the cDPS and RAMPS groups (32.81 vs. 28.88 months, *p* = 0.281; 26.0 vs. 20.0 months, *p* = 0.576, respectively). The 2-year and 5-year overall survival rates for all matched cohorts were 68.5% and 44.8%, respectively, with a median survival time of 48.0 months. Although the median survival time was slightly longer in the cDPS group than in the RAMPS group, the difference was not significant (50.0 vs. 41.0 months, *p* = 0.853). The 2-year and 5-year overall survival rates were comparable between the cDPS and RAMPS groups (71.9% and 44.4% vs. 64.5% and 45.2%, respectively) (Figure 1A). The overall recurrence rate was 73.2%, with no significant difference between both groups (cDPS vs. RAMPS, 75.8% vs. 70.7%, *p* = 0.422). The 2-year and 5-year disease-free survival rates for all matched cohorts were 35.1% and 21.9%, respectively, with no significant difference between the cDPS and RAMPS groups (34.5% and 21.2% vs. 35.4% and 22.3%, respectively). The median DFS time was also similar between the groups (RAMPS: 14.0 months; cDPS: 10.0 months, *p* = 0.929) (Figure 1B).

### 3.5. Prognostic Factor Analysis

In the univariate analysis, several prognostic factors for overall survival (OS) were found to be significant, including age, ASA score, preoperative serum CA19-9, histologic differentiation, lymph node ratio, margin status, adjuvant treatment, PNI, and LVI. However, in the multivariate analysis, only ASA score, preoperative CA19-9, histologic differentiation, margin status, adjuvant treatment, and lymphovascular invasion were identified as independent significant prognostic factors for OS, as shown in Table 4.

Regarding disease-free survival (DFS), in the univariate analysis, preoperative CA19-9, histologic differentiation, T-stage, N-stage, lymph node ratio, adjuvant treatment, PNI, and LVI were identified as significant prognostic factors. However, in the multivariate analysis, only preoperative CA19-9, histologic differentiation, T-stage, adjuvant treatment, and LVI were found to be independent significant prognostic factors for DFS (Table 5).

### 3.6. Subgroup Analysis of Patients who underwent Resection at the Pancreatic Neck

Subgroup analysis was conducted on patients who underwent RAMPS and cDPS at the level of the pancreatic neck. In the univariate analysis, age, sex, histologic differentiation, margin status, and adjuvant treatment were significant prognostic factors for OS. In the multivariate analysis, histologic differentiation, margin statue, and adjuvant treatment were identified as independent significant prognostic factors for OS (Table 6).

Regarding disease-free survival (DFS), in the multivariate analysis, histologic differentiation, T-stage, and adjuvant treatment were found to be independent significant prognostic factors for DFS (Table 7).

## 4. Discussion

RAMPS involves resection of the pancreatic body and tail, as well as an extensive lymph node dissection. Several studies have reported that RAMPS does not increase morbidity and mortality rates [2,9,10,11,12,13]. In our study, the operative outcomes in terms of EBL, transfusion, hospital stay, and the incidence of morbidity, including POPF, PPH, and DGE, were comparable between the groups. In our study, we found that the short-term operative outcomes, including estimated blood loss, transfusion requirements, hospital stay, and the incidence of complications such as postoperative pancreatic fistula, post-pancreatectomy hemorrhage, and delayed gastric emptying, were similar between the RAMPS and cDPS groups. However, the removal of a significant portion of the pancreas during RAMPS can increase the risk of developing diabetes or pancreatic exocrine insufficiency, which may require lifelong enzyme replacement therapy [14,15,16]. The operative time was not significantly different after PSM (Table 2).

Previous studies have demonstrated that RAMPS is associated with a higher number of retrieved lymph nodes and R0 resection rate compared to cDPS [2,3,4,9,11,12,13,17,18]. In contrast, Sham et al. reported lower rates for these parameters in the RAMPS group compared to the cDPS group [10]. In our study, we also observed a significantly higher median number of retrieved lymph nodes in the RAMPS group compared to the cDPS group (15.0 vs. 10.0, *p* = 0.001), but no significant difference in R0 resection rates between the two groups (94.9% in RAMPS vs. 93.9% in cDPS, *p* = 0.756). Although the retrieved LN count was significantly higher in the RAMPS group, there was no evidence that extensive LN dissection can improve survival in patients with pancreatic ductal adenocarcinoma [19]. Several researchers suggested a tailored lymph node dissection extent according to the location of the tumor, based on the results of their research on lymph node metastasis pattern according to the tumor location [20,21]. The N-stage was significantly higher in the RAMPS group, but it was not a significant risk factor in the prognostic factor analysis. In the cDPS group, the number of retrieved lymph nodes (LN) was small, which may have led to downstaging, and this could have influenced the result of the analysis of N-stage as a prognostic factor. Even considering the possibility of downstaging, extensive lymph node resection did not affect survival rate, and adjuvant chemotherapy was a significant prognostic factor. This suggests that adjuvant chemotherapy is more important for improving survival rate than the extent of surgery.

Although RAMPS has been shown to improve retrieved lymph node counts and R0 resection rates, the evidence for its survival benefit remains unclear. Dai et al. reported that the RAMPS group had a higher number of retrieved lymph nodes, longer OS time, and longer DFS time compared to cDPS [3]. However, other studies, including that of Kim et al., Park et al., and Sham et al., found no significant differences in OS and DFS rates between the RAMPS and cDPS groups [4,10,11]. The meta-analysis also showed mixed results, with some authors reporting no significant survival benefit [12,17,18,22], while others reported higher 1-year survival rates in the RAMPS group. In our study, we found no significant difference in OS and DFS between the two groups. By contrast, Dragomir et al. and Zhou et al. reported that the 1-year survival rate was significantly higher in the RAMPS group than in the cDPS group [9,13]. In our study, There was also no significant difference in OS and DFS between the two groups. Translation: In our study, the R0 resection rate, a significant prognostic factor, was similar between the two groups, and there was no difference in survival rate. There was a significant difference in the resection level depending on the location of the tumor, and there was a significant difference in the retrieved LN count, but it did not affect the survival rate. This suggests that if a negative resection margin is obtained, the resection level and retrieved LN count do not affect the survival rate.

Previous studies have demonstrated that margin status, histological grade, lymph node involvement, tumor size, LNR, and CEA are independent prognostic factors in pancreatic ductal adenocarcinoma [1,2,6,23,24,25,26,27]. In the present study, higher ASA score, preoperative CA19-9 > 37 U/mL, poorly/undifferentiated carcinoma, R1 resection, no adjuvant treatment, and lymphovascular invasion were independent poor prognostic factors for OS, and preoperative serum CA19-9 > 37 U/mL, poorly/undifferentiated carcinoma, T3 stage, no adjuvant treatment, and lymphovascular invasion were independent poor prognostic factors for DFS. OS and DFS were not affected by the operation type. Old age and higher ASA scores did not affect DFS. However, they were significant prognostic factors in the univariate analysis for OS, because the patients with old age or poor physical status often could not receive adjuvant chemotherapy.

Our study has limitations, including potential selection bias in this retrospective analysis. To evaluate the value of RAMPS as a standard procedure, the tumor size was included in the covariates for PSM. Therefore large, advanced cases requiring RAMPS to achieve R0 resection could be excluded. R0 resection rate was not significantly different between both groups. Another limitation of this study is that the annual number of cases varies by center, which may result in differences in surgical and pathological expertise. A larger prospective study with longer follow-up is necessary to accurately assess the survival benefit of RAMPS.

## 5. Conclusions

In conclusion, our study demonstrates that RAMPS is a safe and feasible procedure that increases the number of retrieved lymph nodes; however, a significant survival benefit was not observed. Margin status was a significant prognostic factor. Therefore, RAMPS is a treatment option for advanced cases to achieve negative tangential margin rather than a standard procedure for all left-sided pancreatic cancer. Adjuvant treatment remains a significant independent prognostic factor for OS and DFS.

## Figures and Tables

**Figure 1 cancers-16-01546-f001:**
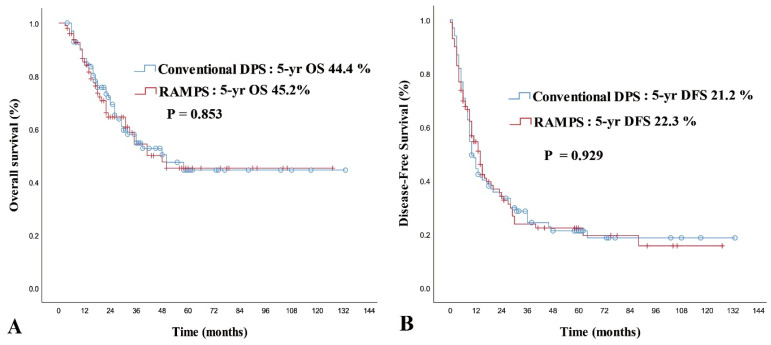
(**A**) Overall survival curve according to the operation type. (**B**) Disease-free survival curve according to the operation type. There were no differences in overall survival or disease-free survival between the conventional DPS and the RAMPS groups.

**Table 1 cancers-16-01546-t001:** Basal characteristics of total and matched cohorts.

	Pre-PSM			Post-PSM		
	cDPS (*n* = 130)	RAMPS (*n* = 203)	*p*-Value	cDPS (*n* = 99)	RAMPS (*n* = 99)	*p*-Value
Age [median (range)] (y)	66.0 (39–86)	64.0 (41–84)	0.503 *	65.0 (39–80)	66.0 (43–81)	0.301 *
Sex [*n* (%)]			0.262			1.000
Male	72 (55.4)	125 (61.6)		53 (53.5)	53 (53.5)	
Female	58 (44.6)	78 (38.4)		46 (46.5)	46 (46.5)	
Approach [*n* (%)]			0.031 †			0.747 †
Open	120 (92.3)	198 (97.5)		93 (93.9)	95 (96.0)	
Laparoscopic	10 (7.7)	5 (2.5)		6 (6.1)	4 (4.0)	
Tumor location [*n* (%)]			<0.001			1.000
Confined to neck/body	58 (44.6)	159 (78.3)		56 (56.6)	56 (56.6)	
Body to tail	19 (14.6)	11 (5.4)		10 (10.1)	10 (10.1)	
Confined to tail	53 (40.8)	33 (16.3)		33 (33.3)	33 (33.3)	
Resection level [*n* (%)]			<0.001			<0.001
At neck	56 (43.1)	203 (100)		42 (42.4)	99 (100)	
Neck~aorta left border	49 (37.7)	0 (0.0)		42 (42.4)	0 (0.0)	
Lateral to aorta left border	25 (19.2)	0 (0.0)		15 (15.1)	0 (0.0)	
BMI (mean ± SD)	23.70 ± 3.288	23.29 ± 2.937	0.234	23.71 ± 3.459	23.41 ± 2.781	0.513
ASA score [*n* (%)]			0.503			0.571
I	14 (10.8)	31 (15.3)		12 (12.1)	14 (14.1)	
II	102 (78.5)	151 (74.4)		78 (78.8)	72 (72.7)	
III	14 (10.8)	21 (10.3)		9 (9.1)	13 (13.1)	
CEA, ng/mL [median (range)]	2.52 (0.20–165.10)	2.20 (0.44–56.92)	0.126 *	2.49 (0.20–165.10)	2.51 (0.44–56.92)	0.734 *
CA19-9, U/mL [median (range)]	59.93 (1.00–11,387.00)	75.01 (1.20–10,028.49)	0.436 *	168.13 (1.80–11,387.00)	244.32 (2.00–6808.37)	0.101 *
Tumor size, cm [median (range)]	3.0 (0.5–9.5)	2.5 (0.3–10.0)	0.282 *	3.0 (1.0–8.0)	3.2 (1.2–8.5)	0.132 *
Neoadjuvant treatment [*n* (%)]			0.606 †			0.246 †
No	125 (96.2)	192 (94.6)		96 (97.0)	99 (100)	
Yes	5 (3.8)	11 (5.4)		3 (3.0)	0 (0.0)	
Adjuvant treatment [*n* (%)]			0.164			0.66
No	54 (41.5)	69 (34.0)		36 (36.4)	39 (39.4)	
Yes	76 (58.5)	134 (66.0)		63 (63.6)	60 (60.6)	

PSM, propensity score matching; DPS, distal pancreatosplenectomy; RAMPS, radical antegrade-modular pancreatosplenctomy; BMI, body mass index; SD, standard deviation; ASA, American Society of Anesthesiologists; CEA, carcinoembryonic antigen; CA, carbohydrate antigen; * Mann–Whitney U test; † Fisher’s exact test.

**Table 2 cancers-16-01546-t002:** Comparison of operative outcomes.

	Pre-PSM			Post-PSM		
	cDPS (*n* = 130)	RAMPS (*n* = 203)	*p*-Value	cDPS (*n* = 99)	RAMPS (*n* = 99)	*p*-Value
LOS [median (range)] (days)	10.0 (5–52)	9.0 (6–152)	0.065 *	10.0 (5–52)	10.0 (7–35)	0.806 *
Approach [*n* (%)]			0.031			0.747
Open	120 (92.3)	198 (97.5)		93 (93.9)	95 (96.0)	
Laparoscopic	10 (7.7)	5 (2.5)		6 (6.1)	4 (4.0)	
Op. time [median (range)] (min)	195 (93–420)	204 (117–494)	0.045 *	195 (98–420)	210 (118–458)	0.305 *
EBL [median (range)] (mL)	250 (30–1600)	300 (50–3000)	0.488 *	250 (50–1600)	250 (50–3000)	0.934 *
Retrieved LN count [median (range)]	10 (0–39)	15 (4–51)	<0.001 *	10.0 (0–36)	15.0 (5–51)	0.001 *
R0 resection [*n* (%)]	120/130 (92.3)	198/203 (97.5)	0.031	93/99 (93.9)	94/99 (94.9)	0.756
Transfusion [*n* (%)]			0.692			>0.99
No	123 (94.6)	194 (95.6)		94 (94.9)	94 (94.9)	
Yes	7 (5.4)	9 (4.4)		5 (5.1)	5 (5.1)	
POPF [*n* (%)]			0.045 †			0.165 †
No or BCL	110 (84.6)	187 (92.1)		85 (85.9)	92 (92.9)	
CR-POPF	20 (15.4)	16 (7.9)		14 (14.1)	7 (7.1)	
DGE [*n* (%)]			0.262			0.261
No	127 (97.7)	199 (98.0)		96 (97.0)	96 (97.0)	
Grade A	0 (0.0)	3 (1.5)		0 (0.0)	2 (2.0)	
Grade B	1 (0.8)	1 (0.5)		1 (1.0)	1 (1.0)	
Grade C	2 (1.5)	0 (0.0)		2 (2.0)	0 (0.0)	
PPH [*n* (%)]			0.22			0.384
No	125 (96.2)	201 (99.0)		95 (96.0)	98 (99.0)	
Grade A	1 (0.8)	0 (0.0)		1 (1.0)	0 (0.0)	
Grade B	3 (2.3)	0 (0.0)		2 (2.0)	0 (0.0)	
Grade C	1 (0.8)	2 (1.0)		1 (1.0)	1 (1.05)	
Chyle leak [*n* (%)]			0.300			0.251
No	126 (96.9)	191 (94.1)		95 (96.0)	90 (90.9)	
Yes	4 (3.1)	12 (5.9)		4 (4.0)	9 (9.1)	
SSI [*n* (%)]			0.966			0.884
No	128 (98.5)	199 (98.0)		97 (98.0)	96 (97.0)	
Superficial	1 (0.8)	3 (1.5)		1 (1.0)	2 (2.0)	
Organ/space	1 (0.8)	1 (0.5)		1 (1.0)	1 (1.0)	
Severe complication [*n* (%)]			0.998			0.663
No	114 (87.7)	178 (87.7)		86 (86.9)	88 (88.9)	
Yes	16 (12.3)	25 (12.3)		13 (13.1)	11 (11.1)	
Recurrence [*n* (%)]			0.251			>0.99
No	37 (28.5)	70 (34.5)		24 (24.2)	24 (24.2)	
Yes	93 (71.5)	133 (65.5)		75 (75.8)	75 (75.8)	
Recurrence pattern [*n* (%)]			0.177			0.507
No	37 (28.5)	70 (34.5)		24 (24.2)	24 (24.2)	
Locoregional	20 (15.4)	17 (8.4)		16 (16.2)	11 (11.1)	
Systemic	61 (46.9)	92 (45.3)		49 (49.5)	51 (51.5)	
Both	12 (9.2)	24 (11.8)		10 (10.1)	13 (13.1)	

PSM, propensity score match; DPS, distal pancreatosplenectomy; RAMPS, radical antegrade modular pancreatosplenectomy; LOS, length of stay; Op, operation; EBL, estimated blood loss; LN, lymph node; POPF, postoperative pancreatic fistula; BCL, biochemical leak; CR-POPF, clinically relevant POPF; DGE, delayed gastric emptying; PPH, post-pancreatectomy hemorrhage; SSI, surgical site infection; * Mann–Whitney U test; † Fisher’s exact test.

**Table 3 cancers-16-01546-t003:** Histopathologic results before and after propensity score matching.

	Pre-PSM			Post-PSM		
	cDPS (*n* = 130)	RAMPS (*n* = 203)	*p*-Value	cDPS (*n* = 99)	RAMPS (*n* = 99)	*p*-Value
Differentiation [*n* (%)]			0.472			0.677
Well	18 (13.8)	20 (9.9)		13 (13.4)	12 (12.1)	
Moderate	82 (63.1)	129 (63.5)		61 (62.9)	68 (68.7)	
Poorly/undifferentiated	30 (23.1)	54 (26.6)		23 (23.7)	19 (19.2)	
T-stage [*n* (%)]			0.208			0.216
T1	30 (23.1)	65 (32.0)		23 (23.2)	19 (45.2)	
T2	72 (55.4)	98 (48.3)		54 (54.5)	47 (47.5)	
T3	28 (21.5)	40 (19.7)		22 (22.2)	33 (33.3)	
Involved LN count [median (range)]	0.0 (0–15)	1.0 (0–22)	0.028	1.0 (0–15)	1.0 (0–17)	0.006
LNR [median (range)]	0.048 (0.0–1.00)	0.057 (0–0.71)	0.383	0.042 (0–1.0)	0.090 (0–0.71)	0.059
N-stage [*n* (%)]			0.083 *			0.024
N0	62 (47.7)	83 (40.9)		44 (44.4)	31 (41.3)	
N1	48 (36.9)	84 (41.4)		38 (38.4)	42 (42.4)	
N2	12 (9.2)	36 (17.7)		12 (12.1)	26 (26.3)	
Nx	8 (6.2)	0 (0.0)		5 (5.1)	0 (0.0)	
Perineural invasion [*n* (%)]			0.360			0.290
PNI−	17 (13.1)	21 (10.3)		13 (13.5)	8 (8.2)	
PNI+	106 (81.5)	180 (88.7)		83 (86.5)	90 (91.8)	
Unknown	7 (5.4)	2 (1.0)		3 (3.0)	1 (1.0)	
Lymphovascular invasion [*n* (%)]			0.376			0.281
LVI−	64 (49.2)	95 (46.8)		44 (44.4)	37 (37.4)	
LVI+	46 (35.4)	85 (41.9)		39 (39.4)	50 (50.5)	
Unknown	20 (15.4)	23 (11.3)		16 (16.2)	12 (42.9)	

PSM, propensity score matching; cDPS, conventional distal pancreatosplenectomy; RAMPS, radical antegrade modular pancreatosplenectomy; LN, lymph node; LNR, lymph node ratio; PNI, perineural invasion; LVI, lymphovascular invasion. * Nx was excluded from the analysis of N-stage.

**Table 4 cancers-16-01546-t004:** Risk factor analysis for overall survival after propensity score matching.

	Univariate					Multivariate	
	*n*	2YSR (%)	5YSR (%)	MST (Months)	*p*-Value	HR (95% CI)	*p*-Value
Age (years)					0.008		
<65	95	75.5	56.1	85.62		1 (Reference)	
65–75	75	66.1	34.3	51.77		1.663 (0.899–3.079)	0.105
>75	28	47.5	34.6	46.71		1.328 (0.511–3.453)	0.561
Sex					0.060		
Male	106	64.2	33.7	61.72			
Female	92	73.4	55.5	80.35			
Operation type					0.853		
Conventional DPS	99	71.9	44.4	72.90		1 (Reference)	
RAMPS	99	64.5	45.2	69.82		1.014 (0.568–1.811)	0.962
ASA physical status					0.047		
1	26	87.6	74.9	105.04		1 (Reference)	
2	150	66.0	41.9	66.87		4.553 (1.081–19.172)	0.039
3	22	63.6	31.9	51.69		5.494 (1.148–26.298)	0.033
CEA					0.079		
≤6 ng/mL	147	69.0	45.9	74.40			
>6 ng/mL	24	65.4	29.3	36.86			
CA19-9					0.039		
≤37 U/mL	49	75.9	57.9	84.98		1 (Reference)	
>37 U/mL	149	65.9	39.6	66.59		2.155 (1.112–4.174)	0.023
Differentiation					<0.001		
Well/moderate	25	74.0	49.8	79.24		1 (Reference)	
Poorly/undifferentiated	42	46.8	25.3	40.58		2.299 (1.290–4.096)	0.005
T-stage					0.143		
T1	42	74.4	58.8	79.15			
T2	101	68.0	37.0	64.39			
T3	55	64.7	42.9	59.61			
N-stage (Nx excluded)					0.254		
N0	75	74.4	53.0	72.89			
N1	80	65.0	37.6	65.97			
N2	38	58.9	26.9	45.57			
Nx	5	80.0	80.0	45.40			
Lymph node ratio					0.016		
<0.2	143	72.1	49.6	78.24		1 (Reference)	
≥0.2	50	55.8	26.0	44.81		1.758 (0.718–4.305)	0.217
Margin status					<0.001		
R0	187	70.9	47.5	75.93		1 (Reference)	
R1	11	31.8	0.0	21.06		4.583 (2.034–10.325)	<0.001
Adjuvant treatment					0.005		
Yes	118	74.8	50.2	79.90		1 (Reference)	
No	80	56.6	35.3	54.20		1.915 (1.112–3.298)	0.019
Perineural invasion					0.032		
No	21	80.7	74.9	99.32		1 (Reference)	
Yes	173	65.9	39.4	67.08		3.423 (0.813–14.415)	0.093
Lymphovascular invasion					0.003		
No	81	80.5	55.8	83.83		1 (Reference)	
Yes	89	63.6	31.7	50.78		2.054 (1.196–3.528)	0.009

YSR, year survival rate; MST, mean survival time; HR, hazard ratio; CI, confidence interval; DPS, distal pancreatosplenectomy; RAMPS, radical antegrade modular pancreatosplenectomy; ASA, American Society of Anesthesiologists; CEA, carcinoembryonic antigen; CA19-9, cancer antigen 19-9.

**Table 5 cancers-16-01546-t005:** Risk factor analysis for disease-free survival after propensity score matching.

	Univariate					Multivariate	
	*n*	2YSR (%)	5YSR (%)	MST (Months)	*p*-Value	HR (95% CI)	*p*-Value
Age (years)					0.290		
<65	95	38.5	28.1	40.71			
65–75	75	32.1	14.5	26.58			
>75	28	29.7	14.8	25.32			
Sex					0.341		
Male	106	32.7	17.1	33.14			
Female	92	37.7	26.7	38.46			
Operation type					0.929		
Conventional DPS	99	34.5	21.2	36.25			
RAMPS	99	35.4	22.3	34.19			
ASA physical status					0.665		
1	26	38.1	29.6	43.63			
2	150	34.8	20.4	33.09			
3	22	33.8	21.1	32.15			
CEA					0.580		
≤6 ng/mL	147	37.0	22.3	37.56			
>6 ng/mL	24	32.5	17.3	23.77			
CA19-9					0.003		
≤37 U/mL	49	48.8	28.4	49.56		1 (Reference)	
>37 U/mL	149	30.6	20.0	30.65		1.808 (1.182–2.767)	0.006
Differentiation					0.002		
Well/moderate	25	39.9	25.0	39.62		1 (Reference)	
Poorly/undifferentiated	42	16.0	0.0	18.48		1.729 (1.122–2.664)	0.013
T-stage					<0.001		
T1	42	57.9	36.5	48.88		1 (Reference)	
T2	101	34.9	20.6	35.52		1.627 (0.975–2.718)	0.063
T3	55	18.0	12.8	20.52		2.611 (1.504–4.532)	0.001
N-stage (Nx excluded)					0.039		
N0	75	41.8	29.2	41.74		1 (Reference)	
N1	80	33.1	16.6	30.07		1.030 (0.621–1.630)	0.899
N2	38	21.3	15.9	21.59		1.255 (0.599–2.627)	0.548
Nx	5	60.0	0.0	24.40		1.905 (0.565–6.420)	0.299
Lymph node ratio					0.012		
<0.2	143	38.4	23.9	40.35		1 (Reference)	
≥0.2	50	24.9	15.6	21.18		0.963 (0.629–1.474)	0.864
Margin status					0.563		
R0	187	34.9	22.1	35.88			
R1	11	36.4	0.0	17.46			
Adjuvant treatment					0.018		
Yes	114	36.7	23.1	37.20		1 (Reference)	
No	75	27.9	17.6	28.40		1.848 (1.257–2.717)	0.002
Perineural invasion					0.013		
No	21	52.4	46.6	58.40		1 (Reference)	
Yes	173	31.8	17.0	30.80		1.740 (0.831–3.647)	0.142
Lymphovascular invasion					0.001		
No	81	46.2	26.8	42.50		1 (Reference)	
Yes	89	22.6	13.4	22.20		1.846 (1.256–2.714)	0.002

YSR, year survival rate; MST, mean survival time; HR, hazard ratio; CI, confidence interval; DPS, distal pancreatosplenectomy; RAMPS, radical antegrade modular pancreatosplenectomy; ASA, American society of anesthesiologists; CEA, carcinoembryonic antigen; CA19-9, cancer antigen 19-9.

**Table 6 cancers-16-01546-t006:** Subgroup analysis: risk factor for overall survival of the patients who underwent RAMPS and cDPS at the level of the pancreatic neck.

	Univariate					Multivariate	
	*n*	2YSR (%)	5YSR (%)	MST (Months)	*p*-Value	HR (95% CI)	*p*-Value
Age (years)					0.003		
<65	52	79.9	63.6	89.93		1 (Reference)	
65–75	40	68.4	26.9	48.49		1.359 (0.606–3.046)	0.457
>75	14	43.0	21.5	35.89		3.153 (0.872–11.395)	0.080
Sex					0.044		
Male	58	65.2	31.5	50.67		1 (Reference)	
Female	48	79.7	58.9	85.39		0.434 (0.181–1.041)	0.062
Operation type					0.351		
Conventional DPS	35	69.7	38.0	60.64		1 (Reference)	
RAMPS	71	73.2	49.5	76.02		0.704 (0.333–1.491)	0.360
ASA physical status					0.158		
1	13	83.9	73.4	91.17			
2	77	72.3	43.4	70.60			
3	16	62.5	30.4	49.15			
CEA					0.070		
≤6 ng/mL	79	71.9	47.3	74.04			
>6 ng/mL	12	64.3	25.7	35.09			
CA19-9					0.403		
≤37 U/mL	27	71.2	53.4	80.42			
>37 U/mL	79	72.5	41.8	59.29			
Differentiation					<0.001		
Well/moderate	83	77.0	52.7	80.10		1 (Reference)	
Poorly/undifferentiated	21	46.4	15.5	34.21		3.437 (1.525–7.744)	0.003
T-stage					0.061		
T1	24	85.7	65.6	87.79			
T2	52	68.3	36.8	61.96			
T3	30	65.0	33.0	54.17			
N-stage (Nx excluded)					0.984		
N0	43	72.0	48.1	67.96			
N1	39	73.2	37.6	67.43			
N2	21	70.0	43.8	60.58			
Nx	3	N/A	N/A	N/A			
Lymph node ratio					0.312		
<0.2	78	73.7	48.2	74.58			
≥0.2	25	65.4	28.6	50.64			
Margin status					0.002		
R0	97	75.3	49.0	75.95		1 (Reference)	
R1	9	40.0	0.0	22.98		8.547 (3.160–23.116)	<0.001
Adjuvant treatment					0.023		
Yes	62	77.7	53.5	80.66		1 (Reference)	
No	44	62.1	32.2	53.71		2.493 (1.162–5.346)	0.019
Perineural invasion					0.117		
No	10	88.9	38.9	62.78			
Yes	93	67.1	37.4	56.85			
Lymphovascular invasion					0.514		
No	44	79.6	47.0	75.63			
Yes	51	65.1	44.8	62.18			

YSR, year survival rate; MST, mean survival time; HR, hazard ratio; CI, confidence interval; DPS, distal pancreatosplenectomy; RAMPS, radical antegrade modular pancreatosplenectomy; ASA, American Society of Anesthesiologists; CEA, carcinoembryonic antigen; CA19-9, cancer antigen 19-9.

**Table 7 cancers-16-01546-t007:** Subgroup analysis: risk factor for disease-free survival of the patients who underwent RAMPS and cDPS at the level of the pancreatic neck.

	Univariate					Multivariate	
	*n*	2YSR (%)	5YSR (%)	MST (Months)	*p*-Value	HR (95% CI)	*p*-Value
Age (years)					0.136		
<65	52	41.9	29.3	43.03			
65–75	40	31.2	9.4	23.67			
>75	14	17.9	17.9	24.48			
Sex					0.203		
Male	58	31.5	13.6	25.24			
Female	48	39.1	27.4	41.49			
Operation type					0.930		
Conventional DPS	35	31.0	18.1	32.21		1 (Reference)	
RAMPS	71	36.8	21.2	34.77		0.696 (0.425–1.142)	0.151
ASA physical status					0.540		
1	13	46.2	30.8	45.31			
2	77	34.9	19.7	33.21			
3	16	25.0	12.5	24.50			
CEA					0.432		
≤6 ng/mL	79	37.0	21.9	36.72			
>6 ng/mL	12	30.0	10.0	19.65			
CA19-9					0.052		
≤37 U/mL	27	39.1	27.4	47.41			
>37 U/mL	79	33.7	17.6	26.84			
Differentiation					0.001		
Well/moderate	83	41.1	24.7	40.13		1 (Reference)	
Poorly/undifferentiated	21	9.5	4.8	13.24		1.958 (1.161–3.303)	0.012
T-stage					<0.001		
T1	24	66.0	44.0	56.76		1 (Reference)	
T2	52	31.5	15.0	31.21		2.193 (1.159–4.152)	0.016
T3	30	15.2	7.6	16.14		3.837 (1.107–2.776)	<0.001
N-stage (Nx excluded)					0.208		
N0	43	37.9	29.5	43.21			
N1	39	35.9	11.3	27.64			
N2	21	27.2	20.4	25.37			
Nx	3	N/A	N/A	N/A			
Lymph node ratio					0.170		
<0.2	78	34.9	22.4	38.87			
≥0.2	25	31.1	15.6	23.51			
Margin status					0.708		
R0	97	35.4	21.8	35.83			
R1	9	29.6	0.0	19.22			
Adjuvant treatment					0.014		
Yes	62	47.5	29.3	46.80		1 (Reference)	
No	44	31.2	18.9	30.21		1.753 (1.107–2.776)	0.017
Perineural invasion					0.029		
No	10	60.0	50.0	61.97		1 (Reference)	
Yes	93	31.1	14.8	26.77		1.448 (0.601–3.489)	0.409
Lymphovascular invasion					0.107		
No	44	44.7	21.1	39.60			
Yes	51	25.7	18.0	25.75			

YSR, year survival rate; MST, mean survival time; HR, hazard ratio; CI, confidence interval; DPS, distal pancreatosplenectomy; RAMPS, radical antegrade modular pancreatosplenectomy; ASA, American Society of Anesthesiologists; CEA, carcinoembryonic antigen; CA19-9, cancer antigen 19-9.

## Data Availability

Data are contained within the article.

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
