# Peer review of "Comparison of Survival Outcomes between Radical Antegrade Modular Pancreatosplenectomy and Conventional Distal Pancreatosplenectomy for Pancreatic Body and Tail Cancer: Korean Multicenter Propensity Score Match Analysis"

_cancers, 2024, doi:10.3390/cancers16081546_

Round 1
Reviewer 1 Report
Comments and Suggestions for Authors
The authors present a retrospective multicentre study on short- and long-term outcomes following either distal pancreatosplenectomy or RAMPS. They included 333 patients and performed a PSM analysis resulting in 99 patients for each group. Criteria for PSM was thoroughly selected between patient and tumor-related factors.
With regards to outcomes there were no big differences to the existing literature showing no real differences between the techniques in the short-term. OS and DFS were also not influenced by the operative procedures but merely by tumor-associated factors which highlight the systemic aspect of this oncologic disease.
1) Why did the authors did not include tumors infiltrating the splenic vessels? About clearing the posterior margins, which also reflect the embryological development of the retroperitoneal area, these tumors might benefit most from the RAMPS technique.
2) Did the authors include only anterior RAMPS?
3) Were all resection lines at the neck, or were tumors of the pancreatic tail resected less extendedly. If there was a difference it would be interesting to see whether there is an impact of RAMPS in tumors of the pancreatic tail.
4) Did the authors also look for the incidence of DM following either RAMOS or cDPS?
5) The number of laparoscopic procedures done by the centers is rather low compared to the international standard. What are the reasons for it?
6) Patients included reflect a yearly case number of <10 resections per year. Center size should also be discussed in the study's limitations since it could also be relevant concerning pathological expertise. In this context it is interesting to see that pos LVI is a negative prognostic factor while N+ is not. I also noticed that in some cases the LN yield was zero. Was it on the surgeon’s discretion (as stated in the methods) or could they be missed by the pathologist? If it was on the surgeons’ discretion, what was the rationale?
7) OS was 0% in R1, while it was 12.8% in DFS. There seems to be a miscalculation.
8) Authors might cancel the last sentence of the abstract as a repetition.
Comments on the Quality of English Language
minor editing required
Author Response
I appreciate your valuable review. I have carefully reviewed your comments and made revisions.
1) Why did the authors did not include tumors infiltrating the splenic vessels? About clearing the posterior margins, which also reflect the embryological development of the retroperitoneal area, these tumors might benefit most from the RAMPS technique.
- Tumors that invaded the splenic vessels were not excluded from this study. However, very advanced cases requiring resection of the celiac axis due to invasion into the celiac axis, or requiring combined organ resection, were excluded as they could not be clearly included in either the cDPS or RAMPS group. Due to the limitations of retrospective studies, we were unable to collect data on splenic vessel invasion from multiple institutions, and therefore could not use the factor for propensity score matching.
2) Did the authors include only anterior RAMPS?
- Our study included 43 cases of posterior RAMPS, but only 7 cases were included after PSM. Therefore, we classified them as a RAMPS group.
3) Were all resection lines at the neck, or were tumors of the pancreatic tail resected less extendedly. If there was a difference it would be interesting to see whether there is an impact of RAMPS in tumors of the pancreatic tail.
- The cDPS group includes cases where resections were performed more laterally than the pancreatic neck, depending on the location of the tumor. Data about the resection level has been added to Table 1.
4) Did the authors also look for the incidence of DM following either RAMOS or cDPS?
- Unfortunately, there was no uniform method for checking the occurrence of diabetes across multiple institutions. While we agree that the occurrence of diabetes is an important factor in deciding the surgical method, the data was unavailable.
5) The number of laparoscopic procedures done by the centers is rather low compared to the international standard. What are the reasons for it?
- In the early period of the study, the frequency of laparoscopic surgery was low because a consensus on the safety of minimally invasive surgery for pancreatic cancer had not been formed. In particular, the number of laparoscopic cases was small due to concerns about the safety of Laparoscopic RAMPS. Recently, the number of laparoscopic surgery has increased in all institutions participating in this study. In future research, the frequency of laparoscopic surgery will appear higher.
6) Patients included reflect a yearly case number of <10 resections per year. Center size should also be discussed in the study's limitations since it could also be relevant concerning pathological expertise. In this context it is interesting to see that pos LVI is a negative prognostic factor while N+ is not. I also noticed that in some cases the LN yield was zero. Was it on the surgeon’s discretion (as stated in the methods) or could they be missed by the pathologist? If it was on the surgeons’ discretion, what was the rationale?
- The limitations related to the size of the center have been added to the discussion section. (line 264) In the cDPS group, the number of retrieved lymph nodes (LN) was small, which may have led to downstaging, and this could have influenced the result of the analysis of N stage as a prognostic factor. The paragraph related to this has been added to the discussion section.(line 224-231) Cases with a lymph node (LN) yield of 0 are those in which LN assessment was not performed in past pathological examinations. During cDPS, lymph nodes around the splenic vessels and at the splenic hilum are resected, so there were no cases where lymph node resection was intentionally not performed.
7) OS was 0% in R1, while it was 12.8% in DFS. There seems to be a miscalculation.
- There was a mistype. I have rechecked the data and made corrections. (table 5)
8) Authors might cancel the last sentence of the abstract as a repetition.
- I have deleted the last sentence of the abstract as your advice.
Reviewer 2 Report
Comments and Suggestions for Authors
This is an exciting study about a topic of interest, particularly for pancreatic surgeons. The paper is scientifically sound, well written, and uses the correct methods, and the results sustain the conclusions. A few issues should be addressed before acceptance:
It is unclear how the decision to perform RAMPS or cDPS was made. Please clarify.
The authors found in the present study that significantly more lymph nodes were retrieved, more positive lymph nodes, and more advanced N stages for RAMPS than cDPS. However, no differences between the groups were observed in disease-free and overall survival. Furthermore, the N stage was not a prognostic factor for disease-free and overall survival. One might consider these as surprising aspects. Please comment.
The Discussion part should be expanded to discuss better the present study's findings, including the prognostic factors.
In line 78, after Strasberg et al., please add the reference.
If there are no benefits for RAMPS over cDPS in terms of recurrence and survival (but potentially impaired long-term functional outcomes), when do the authors consider that RAMPS is indicated?
A native English speaker should revise the manuscript to improve fluency.
Comments on the Quality of English LanguageMinor editing of English language required
Author Response
I appreciate your valuable review. I have carefully reviewed your comments and made revisions.
It is unclear how the decision to perform RAMPS or cDPS was made. Please clarify.
- Initially, the surgical method was chosen by the surgeon at the time of surgery. As described in the research method section, cases that fully complied with Strasberg’s RAMPS inspection based on surgical records and postoperative CT scans were classified as RAMPS. (line 76-85)
The authors found in the present study that significantly more lymph nodes were retrieved, more positive lymph nodes, and more advanced N stages for RAMPS than cDPS. However, no differences between the groups were observed in disease-free and overall survival. Furthermore, the N stage was not a prognostic factor for disease-free and overall survival. One might consider these as surprising aspects. Please comment.
- In the cDPS group, the number of retrieved lymph nodes (LN) was small, which may have led to downstaging, and this could have influenced the result of the analysis of N stage as a prognostic factor. The paragraph related to this has been added to the discussion section. (line 224-231)
The Discussion part should be expanded to discuss better the present study's findings, including the prognostic factors.
- We expanded the discussion part focusing on the findings of our study.
In line 78, after Strasberg et al., please add the reference.
- The reference was added.
If there are no benefits for RAMPS over cDPS in terms of recurrence and survival (but potentially impaired long-term functional outcomes), when do the authors consider that RAMPS is indicated?
- As described in the conclusion, RAMPS is not a standard procedure for all pancreatic body and tail cancers, but rather an option to achieve an R0 margin. Therefore, RAMPS should be performed selectively depending on the extent of the tumor.
A native English speaker should revise the manuscript to improve fluency.
Reviewer 3 Report
Comments and Suggestions for Authors
The article "Comparison of survival outcomes between Radical Antegrade Modular Pancreato-Splenectomy and conventional distal pancreato-splenectomy for pancreatic body and tail cancer: Korean Multicenter Propensity Score Match analysis" is a well-studied multi-centered study with interesting findings. There are few studies in the current topic around the world. This study will serve as another addition highlighting resection using Radical Antegrade Modular Pancreato-Splenectomy carries certain advantages (like higher retrieved lymph node count) over conventional distal pancreato-splenectomy without improving survival outcomes for pancreatic body and tail cancer.
Although the subject is not novel within the literature, but study/research/analysis is well described and conducted. The discussion of the results clarifies all the questions that the applied methodology raises. Also, authors have discussed/acknowledged the limitations of the current study. The available research is clearly presented and discussed, and the conclusion is supported by the evidence presented. The outcome of this present study can be a reference for patient care management. The paper is interesting and fit for the publication.
Author Response
I appreciate your kind comment.
Round 2
Reviewer 1 Report
Comments and Suggestions for Authors
Thank you for addressing all issues raised.
The different resection levels in the cDP group undermine the PSM analysis since the two groups are no more comparable. I would suggest to perform a subgroup analysis (maybe 1:2) only between resections performed at the pancreatic neck.
Comments on the Quality of English Languagenone
Author Response
We performed Propensity Score Matching (PSM) solely based on factors that could be confirmed prior to surgery. Given that age, ASA score, gender, tumor location, tumor size, and serum CA19-9 levels were comparable in both groups, we can say that the selection of groups by PSM was balanced. The different resection level in cDPS is not an imbalance between the groups, but rather can be considered one of the characteristics of cDPS when compared with RAMPS, which has the characteristic of maximal extent. We conducted a subgroup analysis for the cDPS group that underwent resection at the pancreas neck and for RAMPS.
Round 3
Reviewer 1 Report
Comments and Suggestions for Authors
all topics were properly addressed
Comments on the Quality of English Languageall topics were properly addressed